**Comment**

# AI and digital health for childhood cancer care in Ghana

Eric NY Nyarko, Sheila Santa & Patrick Diaba-Nuhoho

Childhood cancer poses a serious challenge to public health in Ghana. Here, the use of AI and digital health tools for childhood cancer care management is explored

Childhood cancers, though relatively rare compared to adult cancers, present a major global health challenge[1]. Each year, about 400,000 children and adolescents (aged 0–14) are newly diagnosed worldwide[2,3]. However, this estimate may fall short due to underreporting and the lack of reliable cancer registries, especially in low and middle-income countries (LMICs)[4]. In the United States and Europe, the incidence of childhood cancer is estimated at around 150–160 cases per million children annually, while reported rates in LMICs range between 30 to 60 cases per million[4]. Globally, the most common types of childhood cancers include leukemia (about 30% of all cases, especially acute lymphoblastic leukemia (ALL)), central nervous system (CNS) tumours (20–25%), lymphomas (Hodgkin and non-Hodgkin lymphoma, 12–15%). Other prevalent types are neuroblastoma, Wilms tumour, retinoblastoma, and bone cancers such as osteosarcoma and Ewing sarcoma[5]. In high-income countries (HICs), leukemia, particularly ALL, and CNS tumours are the leading types[4]. In contrast, Burkitt lymphoma, Kaposi sarcoma (often linked to HIV infection), and retinoblastoma are more common in Sub-Saharan Africa (SSA) and Southeast Asia[6]. In HICs, the 5-year survival rate for childhood cancers has significantly improved, reaching 80–85% due to advancements in diagnostics, treatment (chemotherapy, radiotherapy, and surgery), and supportive care[7]. However, survival rate remains low in many LMICs at approximately around 20–30%[8]. This disparity is often attributed to delayed diagnosis, limited access to treatment, lack of specialised healthcare providers and inadequate follow-up care in LMICs[9]. Ghana is no exception, where childhood cancer poses a serious public health challenge[10]. Although paediatric cancers account for less than 5% of all cancers reported at health facilities in Ghana[11], outcomes are greatly affected by limited healthcare infrastructure, delayed diagnoses, restricted access to essential treatments, financial barriers, and cultural beliefs[12,13]. These challenges highlight the urgent need for innovative solutions to bridge gaps in care delivery and enhance survival outcomes.

Ghana's healthcare landscape is characterized by a significant presence of government-owned facilities, with over 82% (approximately 7,745) of health and medical facilities under public ownership. The remaining 18% are owned by private entities, faith-based organizations, mines, and other quasi-government agencies[14]. The country is divided into 16 regions, with the Ashanti region being the second largest and the largest number of health facilities (18%)[15]. The Greater Accra Region, which is the capital city, comes in second with 14% of the country's health facilities. Together, these two regions account for over a third (35.3%) of Ghana's population[16,17]. On the other hand, Ahafo and Northeast regions reported the least number of health facilities in the country, at 189 and 132, respectively[14]. However,

Ghana is facing a significant shortage of trained healthcare workers, with a deficit of over 87,000. This number is projected to increase to over 113,000 by 2030, based on current population growth rates and the number of healthcare workers required[18]. The existing problem is compounded by the growing migration of health workers to high-income countries which threatens to further weaken health systems in Ghana[19,20]. In response to the critical shortage of healthcare workers in Ghana, the integration of technology, artificial intelligence (AI), and digital health (DH) solutions is increasingly being recognized as a viable strategy to enhance healthcare infrastructure and mitigate workforce deficits. The Ghana Health Service (GHS) has articulated this approach in its policy and strategy on DH 2023–2027, aiming to improve healthcare accessibility, quality, and efficiency through digital innovation[21]. This policy emphasizes the importance of utilizing digital tools to strengthen the health system, particularly in underserved areas. Furthermore, the GHS has endorsed initiatives like the Health Community of West Africa's AI project (HCOWA), which focuses on integrating AI into healthcare delivery to address workforce shortages and improve service provision (https://hcowaa.com/ghana-health-service-backs-hcowas-ai-initiative-for-healthcare-transformation/). These efforts align with global trends where AI and digital technologies are employed to enhance healthcare services, especially in regions facing human resource constraints. With these innovations, the country can improve access to healthcare services, particularly in remote and underserved areas.

AI and DH technologies offer promising solutions to improve childhood cancer outcomes in resource-limited settings like Ghana. AI can enhance early detection, optimise treatment protocols, and support personalized care. For example, machine learning algorithms can analyse complex data such as patient symptoms, diagnostic images, and genomic profiles to help clinicians make accurate, timely diagnoses[22]. Additionally, AI-powered decision support systems provide evidence-based treatment recommendations tailored to the unique needs of each patient, thus improving care precision[23]. Digital health tools, including telemedicine, mobile health (mHealth) platforms, and electronic health records (EHRs) are already in use in several countries[24]. In Ghana where healthcare resources are often concentrated in urban centres, telemedicine can reduce the need for long-distance travel, enabling remote consultations with oncology specialists and expanding access to care for children in rural areas[25]. mHealth applications can also help caregivers manage treatment adherence and monitor symptoms, leading to better long-term outcomes[26]. Thus, the integration of AI and DH technologies has the potential to transform paediatric oncology in Ghana. By utilizing these technologies, Ghana could improve early detection, increase treatment accuracy, and ultimately raise survival rates for children with cancer.

While it is true that there is limited literature on the direct application of AI technologies in childhood cancer care in Africa and LMICs, there are promising examples of AI and DH tools being employed in other medical conditions in these regions. These successes can serve as a foundation for adapting and transferring these technologies to childhood cancer care in

Ghana. Specifically, AI can be applied in four key areas of cancer management: oncologist consultation and patient symptom assessment, diagnosis, treatment, and research data management. Other LMICs have successfully implemented some of these AI-powered solutions in healthcare. For example, the Saathealth mobile AI app (https://medium.com/saathealth-blog) has improved cognitive development and nutritional assessment in children in India[27]. Similarly, the telehealth digital platform, Laura Digital ER (https://fairlac.iadb.org/en/laura-br), has been used to triage and monitor patient symptoms during the COVID-19 pandemic in Brazil[28]. These platforms could be adapted and trained on childhood cancer data in Ghana and, by extension, Africa, to support cancer assessment in children. Furthermore, the "Brilliant Doctor" clinical decision support system has demonstrated positive outcomes in rural Chinese primary-care clinics by suggesting diagnostic alternatives to physicians and reducing medical errors[29]. These innovations could be ported and trained to support cancer management in Ghanaian children.

## Epidemiology and current state of childhood cancer in Ghana

Childhood cancers are a significant public health issue in Ghana particularly in urban centres like Accra and Kumasi where most paediatric oncology services are based[10]. Between 2015 to 2019, the childhood cancer incidence rate in Ghana was 9.36 per 100,000 person-years, with the highest rates observed among children under five and in boys[10]. The most common cancers in Ghanaian children include lymphomas, leukemia, retinoblastoma (eye cancer), and kidney cancers (Wilms tumour); with lymphomas accounting for 30.7% of cases at Korle Bu Teaching Hospital (KBTH), followed by leukemia (18.8%) and retinoblastoma (15.8) (see Fig. 1)[10]. However, national data is limited, and much of the current knowledge comes from major paediatric oncology centres such as KBTH and Komfo Anokye Teaching Hospital (KATH) which handle the majority of cases in the country[10].

Despite advances in diagnosing and treating paediatric cancers in Ghana, a key challenge is the uneven distribution of healthcare resources and expertise[30]. Paediatric cancer care is primarily available at major hospitals, limiting access for children in rural and underserved areas. Additional barriers, including delayed diagnosis due to limited awareness, inadequate referral systems, reliance on traditional medicine, and high treatment abandonment rates compounds the issue. These factors contribute to a childhood cancer survival rate of only 20–30%, significantly lower than global averages[11].

## The potential of AI and digital health in improving childhood cancer care in Ghana

The potential for AI and DH solutions to transform childhood cancer care is particularly relevant in resource-limited settings[31]. While AI-powered tools for paediatric cancer management are largely unavailable in Ghana, technologies like image analysis systems could be adapted to analyse X-rays, magnetic resonance images (MRIs), and histopathological slides with high accuracy. AI can also help make personalised paediatric cancer care more accessible by utilizing global data, especially in settings where resources for genomic profiling are limited. Telemedicine platforms already in use in SSA, such as Gezira family medicine project (GFMP), Réseau en Afrique Francophone pour la Télémédecine (RAFT), Internet Pathology Suite (iPath), the African teledermatology project, the Swinfen charitable trust network and Pan-African E-network, could be adapted for use in Ghana[32,33]. The GFMP proposed an innovative approach for family medicine practitioners in Sudan's Gezira state, integrating information and communication technology (ICT) through training and practice[34]. In Francophone Africa, the RAFT network has been instrumental in educating health professionals and providing medical consultations to rural communities across the region (https://raft.network/presentation). Meanwhile, the iPath has been successfully implemented in Switzerland and is now used globally, including in Africa, for the sharing and discussion of pathological micrographs, as well as the diagnosis of cancers and other diseases[35]. In response to the shortage of dermatologists and skin specialists in developing African countries, the African teledermatology project has been established to facilitate data, knowledge, and skill sharing between African countries and the West, while providing dermatological services to hard-to-reach rural communities (http://africa.telederm.org/). Engineered by Sankalpa Ghose and originally founded by the Swinfen charitable trust network, the Swinfen telemedicine (now OpenTelemed) platform connects medical practitioners and facilities across the globe, specializing in diverse medical specialties (https://www.swinfentelemed.org/ and https://www.opentelemed.org/). While these platforms have undergone significant transformations over the past three years, there is a pressing need to upscale, retool, and develop them into user-

**Fig. 1 | Approximate figures of paediatric cancer cases in Ghana.** The most common cancers in Ghanaian children and includes lymphoma (16%), leukemia (19%), retinoblastoma (30%), and about 10 others (35%).

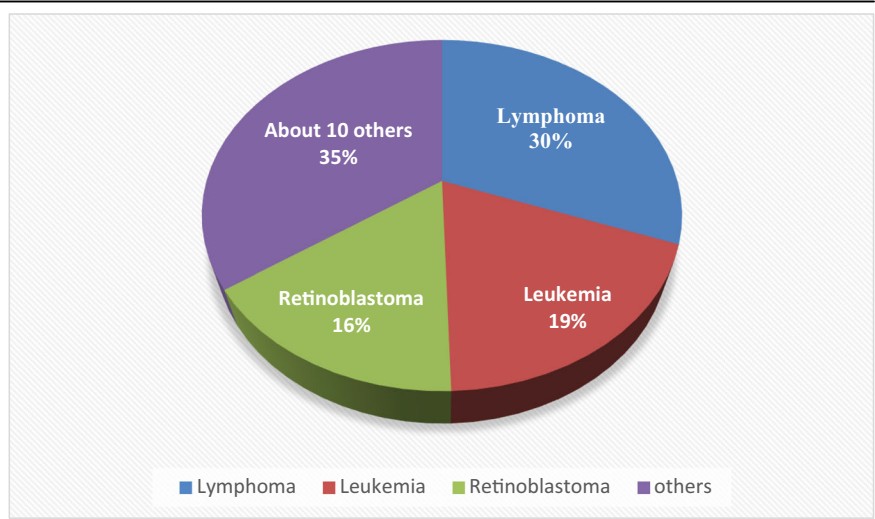

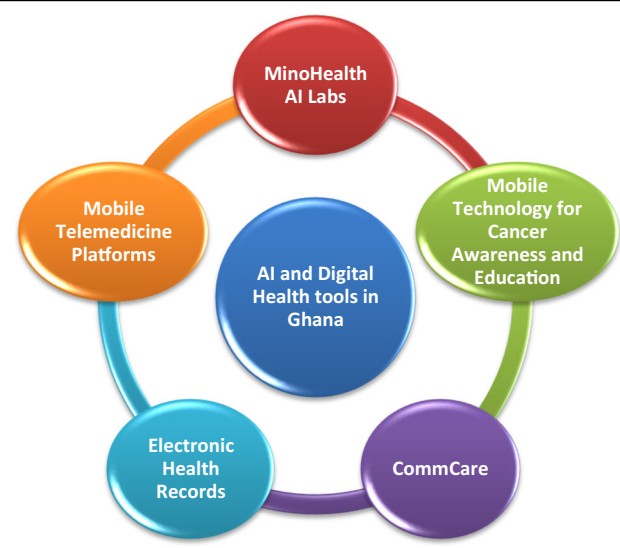

**Fig. 2 | AI and Digital Health tools currently in use in Ghana.** Illustration of AI and digital tools that are integrated to create a unified and efficient system for childhood cancer in Ghana.

friendly apps, websites, USSD platforms, and AI applications for childhood cancer care in Ghana.

With Ghana's high mobile phone penetration, telemedicine could connect rural patients to urban paediatric cancer centres. Furthermore, AI-powered mHealth tools can monitor symptoms, medication adherence, and overall health, enabling timely interventions from healthcare providers and improving treatment adherence[36]. AI can also enhance medical education and training, especially for providers in rural areas. Clinical scenario simulation and real-time decision support could improve early detection and referral rates for paediatric cancer[37]. Current efforts to integrate AI and digital health tools in Ghana are illustrated in Fig. 2. The integration of AI and DH tools in Ghana has the potential to create a unified and efficient system for managing childhood cancer. MinoHealth AI Labs can utilize their expertise in diagnosing childhood cancers, and share this data with various telemedicine platforms. These platforms can then use this data to enable prompt care delivery for the child, and also update the EHRs of all clinics visited by the child. This integrated system can also facilitate the sharing of patient data between different healthcare units, enabling a more comprehensive understanding of the child's medical history. Furthermore, Mobile Technology for Cancer Awareness and Education (MOTECH) can use the data from MinoHealth to create awareness within the community, providing critical information on risk factors and promoting early reporting and prevention. This integrated approach as shown in Fig. 2, can occur in multiple directions, depending on the needs of the patient or population. By fostering interoperability and integration between AI and DH tools, one can create a more effective and efficient system for managing childhood cancer in Ghana.

## MinoHealth AI labs
MinoHealth AI Labs, a leading company founded in 2016 by a Ghanaian, Darlington Akogo, specialises in providing AI solutions for healthcare applications, focusing on areas such as radiology and microscopy[38]. This AI for health care applications, is further driving significant breakthroughs in oncology, regenerative medicine, neuroscience, optometry, epidemiology and nutrition in Ghana. The application of AI to these specialized areas is

achieved through the implementation of biomedical research, clinical research and system developments in collaborations with recognized and established agencies such as the United Nations Global Initiative on AI for Health (GI-AI4H), the Large Language Model AI2 (LLAMA) by META AI. For example, MinoHealth has made significant advancements in the fields of oncology and radiology by developing AI systems that can diagnose 14 thoracic conditions, including pneumonia, fibrosis, hernia, and pleural effusion, through chest X-rays. These systems also assist in breast cancer detection via mammography. Clinical evaluations have shown that these AI models outperform radiologists in diagnosing conditions like cardiomegaly and pleural effusion, achieving higher accuracy rates benefiting healthcare facilities and clinicians in Ghana and over 20 other countries worldwide[38]. In collaboration with Imperial College London, MinoHealth is working on the development of the Lacewing handheld molecular diagnostic device. This innovative lab-on-a-chip technology enables rapid and accurate diagnosis of infectious diseases such as malaria and COVID-19, with results comparable to PCR tests. This technology has the potential to significantly enhance diagnostic capabilities, particularly in resource-limited settings (https://www.ghanaweb.com/GhanaHomePage/NewsArchive/minoHealth-Imperial-College-London-wins-grant-to-undertake-1017046). Mino-Health has also developed AI tools to track and predict the spread of infectious diseases by collecting and analyzing geographic and diagnostic data. This approach helps develop targeted public health interventions. Additionally, the company has developed AI models to analyze dietary patterns and nutritional data, providing personalized recommendations to address malnutrition and related health issues. This platform has the potential to be adapted for detecting childhood cancers and to lead Africa's expansion in AI.

## Mobile technology for cancer awareness and education
Mobile-based platforms are already being used in Ghana to promote cancer awareness, prevention, and education[26,39]. For instance, short message service (SMS) campaigns provide information on cancer symptoms, prevention methods and the importance of early detection, particularly in communities with limited healthcare access[40]. These platforms could be adapted to focus specifically on childhood cancers, educating parents on early symptoms of common paediatric cancers like leukemia and retinoblastoma. SMS campaigns could also include messages about relevant measures to prevent some childhood cancers (e.g., Hepatitis B vaccination to prevent liver cancer) and emphasise the critical role of early detection, helping reduce delays in seeking treatment.

## CommCare
CommCare, developed by Dimagi Inc (USA), has deployed several health solutions in Ghana. For example the Ghana new born screening app for sickle cell, launched in 2018, has screened over 17,000 babies and is used in over 7 centres including KATH and KBTH[41]. Another example is the mobile midwife and nurses' application, part of the MOTECH initiative, which provides targeted pregnancy and childcare information to women and their families, helping community health workers track care for women and newborns[42]. These CommCare platforms could be expanded to support paediatric cancer care.

## Mobile telemedicine platforms
Mobile telemedicine platforms like mHealth Ghana are already in use, enabling virtual consultations and reducing the need for cancer patients to travel to distant healthcare facilities[26]. A study conducted in the Ashanti region of Ghana found that approximately 65% of healthcare professionals reported having access to mHealth tools with nearly all users finding them

beneficial for enhancing healthcare delivery[43]. In 2011, Novartis Foundation worked with local and international partners on a pilot model called the Ghana telemedicine program by using ICT to connect community health workers to medical specialists via 24-hour teleconsultation centres. This allowed doctors, nurses and midwives in the teleconsultation centres coach community health workers and advise on the treatment of patients, helping them manage emergency cases that were beyond their capacity, avoiding unnecessary referrals[44]. Telepathology is also making significant advancements in Ghana. KBTH in Accra has been equipped with digital pathology equipment through partnerships with the American Society for Clinical Pathology (ASCP), Motic USA, and Memorial Sloan Kettering Cancer Centre (MSKCC)[45]. These initiatives allow local health workers to send digital images of processed pathological specimens to specialists for faster cancer diagnoses, particularly in rural areas. Organizations like the ASCP play a crucial role in providing telepathology solutions, including the installation of slide scanners in laboratories across Africa, including Ghana[46]. Such initiatives could be effectively adapted for paediatric oncology to enhance patient management.

### Electronic health records

Recently, the government of Ghana launched several initiatives aimed at integrating EHRs into cancer care to enhance diagnosis, treatment, and management. One key initiative is the digitisation of healthcare, including cancer care, led by the Ministry of Health through the District Health Information Management System 2 (DHIMS2)[47]. This system facilitates the monitoring of national health data, including cases of adult and paediatric cancers. Additionally, KBTH, one of Ghana's largest cancer treatment centres piloted EHRs systems to digitise patient records, streamlining access to information and treatment protocols[48]. These DH and telemedicine platforms in Ghana could be further expanded to include all paediatric cancer care units.

### Challenges of AI and DH in improving childhood cancer care in Ghana

**Limited electricity and internet infrastructure.** Enhancing childhood cancer care in Ghana through AI and DH offers substantial opportunities but also faces significant challenges. One major issue is the lack of reliable internet connectivity and electricity in many regions of Ghana, particularly in rural areas. Effective AI and DH solutions depend on a robust digital infrastructure, which is often inadequate in low-resource settings[49].

**Data availability, regulatory and policy barriers.** AI systems depend on large datasets for effective training and accurate predictions. In Ghana, the lack of centralised, comprehensive medical records and registries, especially in paediatric oncology, poses a significant challenge. The absence of high-quality data on cancer cases, treatment outcomes, and demographics limits the effectiveness of AI[50]. Additionally, DH initiatives necessitate the collection and management of sensitive health data. Ensuring data privacy and security within Ghana's evolving healthcare system is particularly challenging especially given the growing demand for compliance with global standards like General Data Protection Regulation (GDPR)[51,52]. Currently, Ghana lacks comprehensive regulations governing the use of AI in healthcare, which raises concerns about the ethical deployment of these technologies, potential biases in decision-making algorithms, and accountability for AI-driven clinical decisions[53]. While Ghana has taken steps to promote e-health, the policies surrounding DH are still under development, which limits the scalability of AI-driven DH interventions in cancer care[54].

**Access to technology and skills gap.** The deployment and maintenance of AI technology and DH tools can be costly, and many hospitals and healthcare providers in Ghana lack the budget and technological resources needed for full implementation. The financial constraints facing many hospitals and healthcare providers in Ghana are a significant barrier to implementing cutting-edge technologies like AI and DH. Ghana's healthcare budget has been consistently underfunded, with an average allocation of less than $30 per person from 2021 to 2025[55]. These limited resources restrict the adoption of innovative technologies and digital transformation across healthcare facilities, particularly in rural and under-resourced areas[21]. According to estimates from GHS, the country would need to allocate at least $84 per person to provide basic healthcare to every citizen, a target that has yet to be achieved[15]. The decline in investment from governments, donor agencies, and other partners in the Ghanaian health infrastructure, including the ICT and digital health spaces, has exacerbated the cost of health[56]. Hence, only a few private organizations, which often lack the necessary economic and financial capacity, are able to invest in ICT-related healthcare initiatives, including AI and DH in Ghana[57]. Despite the willingness and readiness of most health professionals in Ghana to adopt these technologies, the lack of resources remains a significant challenge[58]. Additionally, a significant number of healthcare professionals lack training in AI and DH tools, which makes it challenging to integrate these innovations into their practices[59].

**Maintenance and technical support.** Maintaining AI systems requires technical expertise, which may not be readily available in Ghana. Breakdowns or bugs in AI-driven digital platforms can disrupt care if there is insufficient support for troubleshooting and repairs[60]. To address these challenges, a multifaceted approach is needed. This includes investing in infrastructure, building human capacity, developing policies, and ensuring equity and accessibility for all children in Ghana.

### Conclusion

**Future perspective.** Digital health applications in Ghana, such as MinoHealth, Commcare, MOTECH, and various telemedicine platforms, serve diverse helpful purposes. However, due to their untrained nature on Ghanaian childhood cancer data and lack of appropriate technology, they are unable to perform to satisfaction. In contrast, AI-powered bots like the Ping An Good Doctor (PAGD, https://www.pagd.net/allPage/aboutUs/47?lang=EN_US)) in China and Tencent Trusted Doctors (TTD) enables the connection of approximately 440,000 certified doctors with more than 10 million patients online, with services varying from online consulting, e-commerce and physical checks (https://www.mobihealthnews.com/news/asia/tencent-backed-online-healthcare-platform-trusted-doctor-secures-250m)[61], enabling earlier detection and preventive actions by patients, parents, and physicians. Specific challenges to Ghana include the lack of comprehensive childhood cancer registries, paucity of ICT infrastructure, skilled ICT personnel, and absence of specific and tailored laws and policies on AI products, use, and regulations. However, these challenges also bring to bear opportunities for collaborations within Ghana, the entire SSA, and with international communities through research, interoperability, technology, culture sharing, and legislation. For instance, initiatives like the international foundation for integrated care in Europe are applying AI to holistically fight childhood cancer, a lesson worth emulating by Ghana and the African Union (https://unica4eu). The development of a more robust model for prognosing and predicting childhood cancer, as reported in a meta-analysis and systematic reviews of publications across the west, can be utilized to develop a more accurate AI

prediction of childhood cancer based on genetic, environmental, and other risk factors[62]. Additionally, AI tools like Paige.ai (https://www.paige.ai/), widely applied in high-income countries like the USA, have the potential to be repurposed to improve childhood cancer diagnosis in Ghana. International collaborations can facilitate the sharing of knowledge, technology, and best practices, enabling Ghana to influence the expertise of global partners and accelerate the development of effective AI-powered solutions for childhood cancer diagnosis and management.

Moreover, the integration of AI and DH technologies into childhood cancer care in Ghana shows great promise, particularly considering the existing challenges in healthcare delivery, infrastructure, and access. AI powered tools can create predictive models to analyse genetic, environmental, and familial risk factors, enabling earlier identification of children at higher risk for developing cancer. Additionally, AI also has the potential to enhance personalised medicine by tailoring treatments to the unique characteristics of each child's cancer type. By mining genetic data, clinical histories, and real-time patient information, AI can generate customised treatment plans that can optimise outcomes for childhood cancers. Likewise, AI can analyse extensive datasets to identify existing drugs that may be repurposed for specific types of childhood cancers for specific populations, potentially accelerating treatment timelines and reducing costs. In low-resource settings like Ghana, effective resource utilisation is crucial. Thus, AI could optimise resource allocation, particularly in supply chain management, ensuring that cancer treatment drugs and medical supplies are efficiently managed and distributed to prevent shortages. Similarly, AI-powered tools can help deploy healthcare personnel based on real-time patient demand, ensuring that oncologists, nurses, and specialists are available where they are most needed. In cancer research, AI can contribute significantly by analysing large datasets from patients across the country. Machine learning techniques and algorithms can identify patterns in genetic data, patient demographics, treatment responses, and long-term outcomes that might not be apparent to human researchers. This can help identify which children might benefit from clinical trials or emerging therapies, helping to accelerate access to new treatments. Finally, effective implementation of DH technologies in resource-limited settings requires prioritising user-centered design that is integrated with the existing ecosystem, planned for scalability and sustainability through data-driven reliance that utilises open standards and innovation. It also requires building upon and improving existing solutions to ensure data privacy and security through collaborative partnerships[63].

AI can assist in early diagnosis, while DH platforms facilitate remote consultations and continuous monitoring, effectively overcoming geographical barriers that limit access to specialised care in rural areas. Furthermore, AI-powered tools can support healthcare professionals in treatment planning and optimising resource allocation in under-resourced settings. However, the successful implementation of these technologies requires addressing challenges such as data privacy, the digital infrastructure gap, and training healthcare providers to use AI systems effectively. Collaboration among the government, health institutions, and international organisations is crucial for investing in digital infrastructure, capacity building, and policy development. With these concerted efforts, AI and DH can significantly enhance childhood cancer care in Ghana.

## Data availability
Source data for Figs. 1 and 2 can be found in the manuscript. All data are included in the manuscript.

**Eric NY Nyarko** (ID)[1,5], **Sheila Santa**[2,5] & **Patrick Diaba-Nuhoho** (ID)[3,4] ✉
[1]Department of Chemical Pathology, University of Ghana Medical School, Accra, Ghana. [2]Department of Medical Laboratory Sciences, School of Biomedical and Allied Health Sciences, University of Ghana, Accra, Ghana. [3]Institute for Pharmacology and Toxicology, Faculty of Medicine and University Hospital Carl Gustav Carus at TUD, Dresden, Germany. [4]Department of Paediatric and Adolescent Medicine, Paediatric Haematology and Oncology, University Hospital Münster, Münster, Germany. [5]These authors contributed equally: Eric NY Nyarko, Sheila Santa. ✉e-mail: patrick.diaba-nuhoho@tu-dresden.de

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

## Author contributions

E.N.Y.N., S.S., and P.D.-N. Conceptualised and wrote the original draft of the manuscript, S.S. designed the figures, E.N.Y.N., S.S., and P.D.-N. critically reviewed and edited the manuscript. All authors reviewed and approved the final manuscript.

## Funding

## Competing interests

The authors declare no competing interests.
