## [Transparent Peer Review file · Communications Medicine]

AI and Digital Health for Childhood Cancer Care in Ghana

Corresponding Author: Dr Patrick Diaba-Nuhoho

Version 0:

Reviewer comments:

Reviewer #1

(Remarks to the Author)

Thank you for the opportunity to review this article.

This article addresses an important and timely topic, with potential to contribute significantly to the field of childhood cancer care innovation in Ghana. The ideas are compelling, but the paper would benefit from incorporation of additional facts and evidence, which would make the article stronger.

With amendments addressing the points outlined below, the manuscript could achieve higher informational value with accuracy, and clarity, making it suitable for publication.

Points for Improvement:

1. While the manuscript focuses on Ghana, there is limited information situating Ghana's efforts within the broader context of Africa and the global landscape of AI and digital health technologies for childhood cancer care. For example, section introduction can include general information about how AI and digital health tools are being employed for childhood cancer care across Africa and other countries.
2. In the section of discussion and future perspective, consider including a broader comparison of AI application in Ghana and Africa with those in Europa, Asia, and other regions of the world, focusing on both middle-income, and high-income countries. Highlight Ghana's specific challenges, in addition to general ones, such as the lack of data availability, in childhood cancer (as mentioned in section 4.2), and explore lessons that can be learnt from other countries. Identify opportunities, including international collaborations.
3. Figure 2, all five components outlined are connected. A short description should be provided to explain how these components synergize.
4. This article covers many aspects of AI and digital health tools in childhood cancer in Ghana, however certain points would benefit from further elaboration and support from references:
 - a. citations should be accurate and precise. For example, in line 30, 'Childhood cancers, though relatively rare compared to adult cancers, present a major global health 29 challenge¹. Each year, about 400,000 children (aged 0-14) are newly diagnosed worldwide.' Upon reviewing reference 1, it appears that the figure of 400,000 children is from 2017 and does not represent an annual statistic.
 - b. In Line 93-96, several projects in Africa are mentioned. It would be helpful to provide a brief description of each project along with respective references or links to their websites.
 - c. In Line 109: the statement 'This AI for health care applications, is further driving significant breakthroughs in oncology, regenerative medicine, neuroscience, optometry, epidemiology and nutrition in Ghana. ', needs further elaboration and supported by references.
 - d. in line 181, the statement 'many hospitals and healthcare providers in Ghana lack the budget and technological resources needed for full implementation' would be more convincing if support by specific data.

Reviewer #2

(Remarks to the Author)

This is an important topic germane to all LMICs with a capacity to improve cancer care in children in those regions.

The authors with a focus on Ghana need to provide more detailed data on the 'medical topography' of the country vis-a-vis the distribution of the workforce and infrastructure involved in the care of pediatric cancers and how the observed disparities can specifically be closed by utilizing AI and digital health.

It is also interesting to note that pediatric brain tumours was not in the top two causes of pediatric cancers in Ghana as opposed to other countries. What are the thoughts of the authors on this?

The authors listed some of the already utilized AI and digital tools in Ghana referencing other disease conditions, they should discuss how these tools can be specifically directed at pediatric cancers.

Version 1:

Reviewer comments:

Reviewer #1

(Remarks to the Author)

Thank you for your revision. I can see that you have carefully addressed all the points I raised earlier. The changes have strengthened the clarity and quality of the manuscript and I have no further comments.

Reviewer #2

(Remarks to the Author)

The authors have sufficiently improved the manuscript.

Reviewer 1

Thank you for the opportunity to review this article. This article addresses an important and timely topic, with potential to contribute significantly to the field of childhood cancer care innovation in Ghana. The ideas are compelling, but the paper would benefit from incorporation of additional facts and evidence, which would make the article stronger. With amendments addressing the points outlined below, the manuscript could achieve higher informational value with accuracy, and clarity, making it suitable for publication.

1. While the manuscript focuses on Ghana, there is limited information situating Ghana's efforts within the broader context of Africa and the global landscape of AI and digital health technologies for childhood cancer care. For example, section introduction can include general information about how AI and digital health tools are being employed for childhood cancer care across Africa and other countries.

Thank you for this kind suggestion. The introduction has been updated in the following way: "While it is true that there is limited literature on the direct application of AI technologies in childhood cancer care in Africa and Low- and Middle-Income Countries (LMICs), there are promising examples of AI and digital health tools being employed in other medical conditions in these regions. These successes can serve as a foundation for adapting and transferring these technologies to childhood cancer care in Ghana. Specifically, AI can be applied in four key areas of cancer management: oncologist consultation and patient symptom assessment, diagnosis, treatment, and research data management. Other LMICs have successfully implemented some of these AI-powered solutions in healthcare. For example, the Saathealth mobile AI app (<https://medium.com/saathealth-blog>) has improved cognitive development and nutritional assessment in children in India (Ganju et al., 2021). Similarly, the telehealth digital platform, Laura Digital ER (<https://fairlac.iadb.org/en/laura-br>), has been used to triage and monitor patient symptoms during the COVID-19 pandemic in Brazil (Morales et al., 2021). These platforms could be adapted and trained on childhood cancer data in Ghana and, by extension, Africa, to support cancer assessment in children. Furthermore, the "Brilliant Doctor" clinical decision support system has demonstrated positive outcomes in rural Chinese primary-care clinics by suggesting diagnostic alternatives to physicians and reducing medical errors (Wang et al., 2021). These innovations could be ported and trained to support cancer management in Ghanaian children".

2. In the section of discussion and future perspective,
 - a. Consider including a broader comparison of AI application in Ghana and Africa with those in Europe, Asia, and other regions of the world, focusing on both middle-income and high-income countries.

Thank you for this important suggestion. The section is updated: "Digital health applications in Ghana, such as MinoHealth, Commcare, MOTECH, and various telemedicine platforms, serve diverse helpful purposes. However, due to their untrained nature on Ghanaian childhood cancer

data and lack of appropriate technology, they are unable to perform to satisfaction. In contrast, AI-powered bots like the Ping An Good Doctor (PAGD, https://www.pagd.net/allPage/aboutUs/47?lang=EN_US) in China and Tencent Trusted Doctors (TTD) enables the connection of approximately 440,000 certified doctors with more than 10 million patients online, with services varying from online consulting, e-commerce and physical checks (<https://www.mobihealthnews.com/news/asia/tencent-backed-online-healthcare-platform-trusted-doctor-secures-250m>) (Jiang et al., 2021), enabling earlier detection and preventive actions by patients, parents, and physicians.

b. Highlight Ghana’s specific challenges, in addition to general ones, such as the lack of data availability, in childhood cancer, and explore lessons that can be learnt from other countries.

Thank you. The following sentence are added to the section: “Specific challenges to Ghana include the lack of comprehensive childhood cancer registries, paucity of ICT infrastructure, skilled ICT personnel, and absence of specific and tailored laws and policies on AI products, use, and regulations. However, these challenges also bring to bear opportunities for collaborations within Ghana, the entire SSA, and with international communities through research, interoperability, technology, culture sharing, and legislation. For instance, initiatives like the international foundation for integrated care in Europe are applying AI to holistically fight childhood cancer, a lesson worth emulating by Ghana and the African Union (<https://unica4eu>)”.

c. Identify opportunities, including international collaborations.

Thank you for your suggestion. The section is updated: “The development of a more robust model for prognosing and predicting childhood cancer, as reported in a meta-analysis and systematic reviews of publications across the west, can be utilized to develop a more accurate AI prediction of childhood cancer based on genetic, environmental, and other risk factors (Varga et al., 2024). Additionally, AI tools like Paige.ai (<https://www.paige.ai/>), widely applied in high-income countries like the USA, have the potential to be repurposed to improve childhood cancer diagnosis in Ghana. International collaborations can facilitate the sharing of knowledge, technology, and best practices, enabling Ghana to influence the expertise of global partners and accelerate the development of effective AI-powered solutions for childhood cancer diagnosis and management”.

3. Figure 2, all five components outlined are connected. A short description should be provided to explain how these components synergize.

Thank you very much. We added a short description to Figure 2: “The integration of AI and digital health (DH) tools in Ghana has the potential to create a unified and efficient system for managing childhood cancer. MinoHealth AI Labs can utilize their expertise in diagnosing childhood cancers, and share this data with various telemedicine platforms. These platforms can then use this data to enable prompt care delivery for the child, and also update the electronic health records (EHRs) of

all clinics visited by the child. This integrated system can also facilitate the sharing of patient data between different healthcare units, enabling a more comprehensive understanding of the child's medical history. Furthermore, Mobile Technology for Cancer Awareness and Education (MOTEC) can use the data from MinoHealth to create awareness within the community, providing critical information on risk factors and promoting early reporting and prevention. This integrated approach as shown in Figure 2, can occur in multiple directions, depending on the needs of the patient or population. By fostering interoperability and integration between AI and DH tools, one can create a more effective and efficient system for managing childhood cancer in Ghana”.

4. This article covers many aspects of AI and digital health tools in childhood cancer in Ghana; however certain points would benefit from further elaboration and support from references:
 - a. citations should be accurate and precise. For example, in line 30, ‘Childhood cancers, though relatively rare compared to adult cancers, present a major global health 29 challenge¹. Each year, about 400,000 children (aged 0-14) are newly diagnosed worldwide.’ Upon reviewing reference 1, it appears that the figure of 400,000 children is from 2017 and does not represent an annual statistic.

Thank you for pointing this out. The statement is updated with the latest reference: “Each year, about 400,000 children and adolescents (aged 0-14) are newly diagnosed worldwide (Sung et al., 2021; The Lancet Diabetes, 2022)”.

- b. In Line 93-96, several projects in Africa are mentioned. It would be helpful to provide a brief description of each project along with respective references or links to their websites.

Sure! A brief description of the several projects mentioned are provided with their respective link and references: “The Gezira Family Medicine Project (GFMP) proposed an innovative approach for family medicine practitioners in Sudan’s Gezira state, integrating ICT through training and practice (Mohamed et al., 2014). In Francophone Africa, the Réseau en Afrique Francophone pour la Télémédecine (RAFT) network has been instrumental in educating health professionals and providing medical consultations to rural communities across the region (<https://raft.network/presentation>). Meanwhile, the Internet Pathology Suite (iPath) has been successfully implemented in Switzerland and is now used globally, including in Africa, for the sharing and discussion of pathological micrographs, as well as the diagnosis of cancers and other diseases (Brauchli & Oberholzer, 2005). In response to the shortage of dermatologists and skin specialists in developing African countries, the African teledermatology project has been established to facilitate data, knowledge, and skill sharing between African countries and the West, while providing dermatological services to hard-to-reach rural communities (<http://africa.telederm.org/>). Engineered by Sankalpa Ghose and originally founded by the Swinfen charitable trust network, the Swinfen telemedicine (now OpenTelemed) platform connects medical practitioners and facilities across the globe, specializing in diverse medical specialties (<https://www.swinfentelemed.org/> and <https://www.opentelemed.org/>). While these platforms

have undergone significant transformations over the past three years, there is a pressing need to upscale, retool, and develop them into user-friendly apps, websites, USSD platforms, and AI applications for childhood cancer care in Ghana”.

c. In Line 109: the statement ‘ This AI for health care applications, is further driving significant breakthroughs in oncology, regenerative medicine, neuroscience, optometry, epidemiology and nutrition in Ghana, with regards to diagnosis, monitoring and management of these diseases, needs further elaboration and supported by references.

Thank you for the suggestion. We have elaborated and supported our statements with references: “The application of AI to these specialized areas is achieved through the implementation of biomedical research, clinical research and system developments in collaborations with recognized and established agencies such as the United Nations Global Initiative on AI for Health (GI-AI4H), the Large Language Model AI2 (LLAMA) by META AI (<https://minohealth.ai/>). For example, MinoHealth has made significant advancements in the fields of oncology and radiology by developing AI systems that can diagnose 14 thoracic conditions, including pneumonia, fibrosis, hernia, and pleural effusion, through chest X-rays. These systems also assist in breast cancer detection via mammography. Clinical evaluations have shown that these AI models outperform radiologists in diagnosing conditions like cardiomegaly and pleural effusion, achieving higher accuracy rates benefiting healthcare facilities and clinicians in Ghana and over 20 other countries worldwide (Akogo et al., 2022). In collaboration with Imperial College London, MinoHealth is working on the development of the Lacewing handheld molecular diagnostic device. This innovative lab-on-a-chip technology enables rapid and accurate diagnosis of infectious diseases such as malaria and COVID-19, with results comparable to PCR tests. This technology has the potential to significantly enhance diagnostic capabilities, particularly in resource-limited settings (<https://www.ghanaweb.com/GhanaHomePage/NewsArchive/minoHealth-Imperial-College-London-wins-grant-to-undertake-1017046>). MinoHealth has also developed AI tools to track and predict the spread of infectious diseases by collecting and analyzing geographic and diagnostic data. This approach helps develop targeted public health interventions. Additionally, the company has developed AI models to analyze dietary patterns and nutritional data, providing personalized recommendations to address malnutrition and related health issues”.

d. in line 181, the statement ‘many hospitals and healthcare providers in Ghana lack the budget and technological resources needed for full implementation’ would be more convincing if support by specific data.

Thank you for the suggestion. Our statement has been updated with supporting literature and references: “The financial constraints facing many hospitals and healthcare providers in Ghana are a significant barrier to implementing cutting-edge technologies like AI and digital health. Ghana's healthcare budget has been consistently underfunded, with an average allocation of less than \$30 per person from 2021 to 2025 (WHO, 2025). These limited resources restrict the adoption of

innovative technologies and digital transformation across healthcare facilities, particularly in rural and under-resourced areas (GHS, 2023b). According to estimates from Ghana Health Service, the country would need to allocate at least \$84 per person to provide basic healthcare to every citizen, a target that has yet to be achieved (GHS, 2023a). The decline in investment from governments, donor agencies, and other partners in the Ghanaian health infrastructure, including the ICT and digital health spaces, has exacerbated the cost of health (MOH, 2021). Hence, only a few private organizations, which often lack the necessary economic and financial capacity, are able to invest in ICT-related healthcare initiatives, including AI and digital health in Ghana (WHO, 2021). Despite the willingness and readiness of most health professionals in Ghana to adopt these technologies, the lack of resources remains a significant challenge (Mensah et al., 2023)”.

Reviewer 2

This is an important topic germane to all LMICs with a capacity to improve cancer care in children in those regions. The authors with a focus on Ghana need to provide more detailed data on the 'medical topography' of the country vis-a-vis the distribution of the workforce and infrastructure involved in the care of pediatric cancers and how the observed disparities can specifically be closed by utilizing AI and digital health.

1. The authors with a focus on Ghana need to provide more detailed data on the 'medical topography' of the country vis-a-vis the distribution of the workforce and infrastructure involved in the care of pediatric cancers and how the observed disparities can specifically be closed by utilizing AI and digital health.

Thank you for the kind suggestion. The manuscript is updated with this additional information: "Ghana's healthcare landscape is characterized by a significant presence of government-owned facilities, with over 82% (approximately 7,745) of health and medical facilities under public ownership. The remaining 18% are owned by private entities, faith-based organizations, mines, and other quasi-government agencies (Sasu, 2024). The country is divided into 16 regions, with the Ashanti region being the second largest and the largest number of health facilities (18%) (GHS, 2023a). The Greater Accra Region, which is the capital city, comes in second with 14% of the country's health facilities. Together, these two regions account for over a third (35.3%) of Ghana's population (GSS, 2023; Sasu, 2022). On the other hand, Ahafo and Northeast reported the least number of health facilities in the country, at 189 and 132, respectively (Sasu, 2024). However, Ghana is facing a significant shortage of trained healthcare workers, with a deficit of over 87,000. This number is projected to increase to over 113,000 by 2030, based on current population growth rates and the number of healthcare workers required (Asamani et al., 2021). The existing problem is compounded by the growing migration of health workers to high-income countries which threatens to further weaken health systems in Ghana (Gulland, 2013; WHO, 2023). In response to the critical shortage of healthcare workers in Ghana, the integration of technology, artificial intelligence (AI), and digital health solutions is increasingly being recognized as a viable strategy to enhance healthcare infrastructure and mitigate workforce deficits. The Ghana health service has articulated this approach in its policy and strategy on digital health 2023-2027, aiming to improve healthcare accessibility, quality, and efficiency through digital innovation (GHS, 2023b). This policy emphasizes the importance of utilizing digital tools to strengthen the health system, particularly in underserved areas. Furthermore, the Ghana Health Service has endorsed initiatives like the Health Community of West Africa's AI project (HCOWA), which focuses on integrating AI into healthcare delivery to address workforce shortages and improve service provision (<https://hcowaa.com/ghana-health-service-backs-hcowas-ai-initiative-for-healthcare-transformation/>). These efforts align with global trends where AI and digital technologies are employed to enhance healthcare services, especially in regions facing human resource constraints. With these innovations, the country can improve access to healthcare services, particularly in remote and underserved areas".

2. It is also interesting to note that pediatric brain tumours were not in the top two causes of pediatric cancers in Ghana as opposed to other countries. What are the thoughts of the authors on this?

We appreciate the reviewer's observation that paediatric brain tumours were not among the top two causes of paediatric cancers in Ghana, as opposed to other countries. Upon reviewing literature, we noticed that the prevalence of paediatric brain tumours in Ghana is indeed lower compared to other countries. However, one study in Ghana reported brain tumour as one of the highest mortalities of all paediatric cases in females and males (Wiredu & Armah, 2006). They did a 10-year retrospective review (1991-2000) of autopsy records from all the wards of the Korle-Bu Teaching Hospital (KBTH) and obtained data for 375 children (< 15 years). In contrast, a recent retrospective study (2015-2019) by Owusu et al. (2023) conducted in the two main paediatric cancer referral centres in Ghana: KBTH and Komfo Anokye Hospital (KATH) identified brain tumour as generally low in paediatric cancer across the regions of Ghana. In our opinion, the types of cancers that occur in a population are influenced by a combination of genetic and environmental risk factors (Stefan et al., 2017). This could be a major factor for these differences. Aside from the general lack of awareness about childhood cancer, delayed health seeking, and limited access to health services, which can lead to late, missed, or no diagnosis, the high associated cost of medical care and inadequate diagnostic services further worsen the challenges in pediatric cancer management in Ghana. This results in low treatment adherence, insufficiently trained health workers and unaffordable costs. Perhaps, influencing the low incidence of brain tumours and other cancer types.

3. The authors listed some of the already utilized AI and digital tools in Ghana referencing other disease conditions, they should discuss how these tools can be specifically directed at pediatric cancers.

Thank you for your suggestion. Applications of these tools to paediatric cancers have been discussed under these sections (“introduction”, “the potential of AI and digital health in improving childhood cancer care in Ghana” and “future perspective and opportunities”).

We would like to sincerely thank you for your kind suggestions, which has substantially improved the quality of our manuscript.

References

- Akogo, D., Sarkodie, B. D., Samori, I. A., Jimah, B. B., Anim, D. A., & Mensah, Y. B. (2022). Minohealth. ai: A clinical evaluation of deep learning systems for the diagnosis of pleural effusion and cardiomegaly in Ghana, Vietnam and the United States of America. *arXiv preprint arXiv:2211.00644*.
- Asamani, J. A., Christmals, C. D., & Reitsma, G. M. (2021). Modelling the supply and need for health professionals for primary health care in Ghana: Implications for health professions education and employment planning. *PLoS One*, *16*(9), e0257957. <https://doi.org/10.1371/journal.pone.0257957>
- Brauchli, K., & Oberholzer, M. (2005). The iPath telemedicine platform. *J Telemed Telecare*, *11 Suppl 2*, S3-7. <https://doi.org/10.1258/135763305775124795>
- Ganju, A., Satyan, S., Tanna, V., & Menezes, S. R. (2021). AI for Improving Children's Health: A Community Case Study. *Front Artif Intell*, *3*, 544972. <https://doi.org/10.3389/frai.2020.544972>
- GHS. (2023a). *Ghana harmonized health facility assesement 2022-2023*. Ghana Health Service (GHS). [https://ghs.gov.gh/wp-content/uploads/2023/12/National Harmonised Health Facility Assessment Report GHHFA Signed.pdf](https://ghs.gov.gh/wp-content/uploads/2023/12/National_Harmonised_Health_Facility_Assessment_Report_GHHFA_Signed.pdf). Accessed 07/05/2025.
- GHS. (2023b). *Policy and strategy on digital health 2023-2027*. Ghana Health Service (GHS). [https://ghs.gov.gh/wp-content/uploads/2023/04/POLICY & STRATEGY 2023-2027](https://ghs.gov.gh/wp-content/uploads/2023/04/POLICY_%20STRATEGY_2023-2027.pdf). Accessed 07/05/2025.
- GSS. (2023). *2021 Population and housing census*. Ghana Statistical Service (GSS). <https://census2021.statsghana.gov.gh/resources.php?readpage=MjQwMTYxOTI1NC4yNjE1&Download-Census-Data>. Accessed 07/05/2025.
- Gulland, A. (2013). Shortage of health workers is set to double, says WHO. *BMJ*, *347*, f6804. <https://doi.org/10.1136/bmj.f6804>
- Jiang, X., Xie, H., Tang, R., Du, Y., Li, T., Gao, J., Xu, X., Jiang, S., Zhao, T., Zhao, W., Sun, X., Hu, G., Wu, D., & Xie, G. (2021). Characteristics of online health care services from China's largest online medical platform: cross-sectional survey study. *J Med Internet Res*, *23*(4), e25817. <https://doi.org/10.2196/25817>
- Mensah, N. K., Adzakah, G., Kissi, J., Boadu, R. O., Lasim, O. U., Oyenike, M. K., Bart-Plange, A., Dalaba, M. A., & Sukums, F. (2023). Health professional's readiness and factors associated with telemedicine implementation and use in selected health facilities in Ghana. *Heliyon*, *9*(3), e14501. <https://doi.org/10.1016/j.heliyon.2023.e14501>
- MOH. (2021). *Health sector medium term development plan (2022–2025)*. Ministry of Health (MOH). https://www.moh.gov.gh/wp-content/uploads/2022/08/HSM TDP_2022-2025.docx14.pdf. Accessed 07/05/2025.
- Mohamed, K. G., Hunskaar, S., Abdelrahman, S. H., & Malik, E. M. (2014). Scaling up family medicine training in Gezira, Sudan - a 2-year in-service master programme using modern information and communication technology: a survey study. *Hum Resour Health*, *12*(1), 3. <https://doi.org/10.1186/1478-4491-12-3>
- Morales, H. M. P., Guedes, M., Silva, J. S., & Massuda, A. (2021). COVID-19 in Brazil-preliminary analysis of response supported by artificial intelligence in municipalities. *Front Digit Health*, *3*, 648585. <https://doi.org/10.3389/fdgth.2021.648585>
- Owusu, W. E., Burger, J. R., Lubbe, M. S., Joubert, R., & Cockeran, M. (2023). Incidence patterns of childhood cancer in two tertiary hospitals in Ghana from 2015 to 2019: A retrospective observational study. *Cancer Epidemiol*, *87*, 102470. <https://doi.org/10.1016/j.canep.2023.102470>

- Sasu, D. D. (2022). *Regional population distribution in Ghana in 2010 and 2021*. <https://www.statista.com/statistics/1231207/distribution-of-the-population-in-ghana-by-region/>. Accessed 07/05/2025
- Sasu, D. D. (2024). *Distribution of health facilities in Ghana as of August 2022, by region*. <https://www.statista.com/statistics/1238770/number-of-health-facilities-in-ghana-by-region/>. Accessed 07/05/2025
- Stefan, C., Bray, F., Ferlay, J., Liu, B., & Maxwell Parkin, D. (2017). Cancer of childhood in sub-Saharan Africa. *Ecancermedicalscience*, 11, 755. <https://doi.org/10.3332/ecancer.2017.755>
- Sung, H., Ferlay, J., Siegel, R. L., Laversanne, M., Soerjomataram, I., Jemal, A., & Bray, F. (2021). Global Cancer Statistics 2020: GLOBOCAN estimates of incidence and mortality worldwide for 36 cancers in 185 countries. *CA Cancer J Clin*, 71(3), 209-249. <https://doi.org/10.3322/caac.21660>
- The Lancet Diabetes, E. (2022). Childhood cancer survival: a gap in need of closing. *Lancet Diabetes Endocrinol*, 10(3), 149. [https://doi.org/10.1016/S2213-8587\(22\)00050-X](https://doi.org/10.1016/S2213-8587(22)00050-X)
- Varga, P., Obeidat, M., Mate, V., Koi, T., Kiss-Dala, S., Major, G. S., Timar, A. E., Li, X., Szilagyi, A., Csaki, Z., Engh, M. A., Garami, M., Hegyi, P., Turi, I., & Tuboly, E. (2024). From simple factors to artificial intelligence: evolution of prognosis prediction in childhood cancer: a systematic review and meta-analysis. *EClinicalMedicine*, 78, 102902. <https://doi.org/10.1016/j.eclinm.2024.102902>
- Wang, D., Wang, L., Zhang, Z., Wang, D., Zhu, H., Gao, Y., Fan, X., & Tian, F. (2021). "Brilliant AI Doctor" in rural clinics: challenges in AI-powered Cclinical decision support system deployment Proceedings of the 2021 CHI Conference on Human Factors in Computing Systems, Yokohama, Japan. <https://doi.org/10.1145/3411764.3445432>
- WHO. (2021). *Global strategy on digital health 2020–2025*. World Health Organization (WHO). <https://www.who.int/publications/i/item/9789240020924>. Accessed 07/05/2025.
- WHO. (2023). *WHO health workforce support and safeguards list 2023*. World Health Organization (WHO). <https://www.who.int/publications/i/item/9789240069787>. Accessed 07/05/2025.
- WHO. (2025). *Global health expenditure database*. World Health Organization (WHO). <https://apps.who.int/nha/database>. Accessed 07/05/2025.
- Wiredu, E. K., & Armah, H. B. (2006). Cancer mortality patterns in Ghana: a 10-year review of autopsies and hospital mortality. *BMC Public Health*, 6, 159. <https://doi.org/10.1186/1471-2458-6-159>